# Position: The Term "Machine Unlearning" Is Overused in LLMs

Sangyeon Yoon [* 1]   Yeachan Jun [* 1]   Albert No [1]

## Abstract

Large language models increasingly face demands to "forget" training data, knowledge, or behaviors due to regulatory deletion obligations, copyright/licensing disputes, and safety or product-policy requirements. **This position paper argues that *machine unlearning* is overused as a term in LLM research and should be reserved for dataset-defined deletion: removing the training influence of a precisely specified forget set such that the resulting model is (approximately) indistinguishable from retraining without that data.** We contend that many tasks currently labeled "unlearning" (e.g., refusal for harmful requests, entity/knowledge removal, or targeted suppression) pursue different, often policy-dependent objectives and therefore require different terminology and baselines (e.g., alignment, suppression, editing, obfuscation). We further argue that this confusion is not cosmetic: because papers make different implicit guarantees under the same label, metrics and benchmarks are frequently reused outside their intended scope, rewarding surface-level non-disclosure (e.g., low ROUGE/forget accuracy) even when retraining-equivalence is not tested and derived capabilities remain. We conclude by calling for stricter terminology tied to explicit guarantees and reference models, and for evaluations that match the claimed objective.

## 1. Introduction

Foundation models (Achiam et al., 2023; Liu et al., 2024a; Comanici et al., 2025) are trained on large, heterogeneous corpora assembled under mixed licenses, consents, and contractual constraints. As these models are deployed in regulated and commercial settings, service providers increasingly face requests to *remove* the effect of specific train-

ing data, motivated by privacy deletion obligations (e.g., the right to be forgotten (European Parliament & Council of the European Union)), copyright and licensing disputes (*Tremblay v. OpenAI, Inc.*, 2023; *Kadrey v. Meta Platforms, Inc.*, 2023; Grynbaum & Mac, 2023), and enterprise data-governance requirements (Voigt & Von dem Bussche, 2017). These pressures have sharpened interest in *machine unlearning* as a principled way to remove the influence of selected training data.

In the classical machine learning formulation, *machine unlearning* is a dataset-defined deletion problem. Given a training set $D$ and a precisely specified *forget set* $F \subset D$, the goal is to produce an updated model whose behavior is (approximately) indistinguishable from the counterfactual model obtained by retraining from scratch on $D \setminus F$ (Ginart et al., 2019; Guo et al., 2020; Neel et al., 2021; Ullah et al., 2021; Izzo et al., 2021). This definition fixes both the *target* and the *baseline*: it requires removing the training influence of a concrete subset of data, and it judges success by similarity to a model retrained on $D \setminus F$ (or a principled proxy), rather than by whether the outputs satisfy a chosen policy.

However, in recent LLM research, the word "unlearning" is frequently used for a broader range of objectives that share a high-level motivation ("make the model forget X") but do not match the retraining-based guarantee. Examples include preventing harmful behaviors, suppressing specific knowledge, removing entities, or blocking classes of queries (Li et al., 2024; Jin et al., 2024; Choi et al., 2025). These directions are important in practice, especially for safety and product policy, but they typically target *behavioral constraints* rather than dataset-defined deletion. When these objectives are discussed under the same term as machine unlearning, claims and evaluations become difficult to interpret: readers cannot tell whether a method aims to match retraining on $D \setminus F$, or merely to change what the system says under a particular prompting protocol.

A central reason is that many non-compliance "forgetting" requests are inherently *policy-defined* and application-dependent (Li et al., 2024; Jin et al., 2024; Luo et al., 2025). For instance, "forget harmful behavior" (e.g., bomb-making assistance) requires choosing a boundary: should the system block only step-by-step weaponization instructions, or also broadly relevant chemistry knowledge? Likewise, "forget

---
*Equal contribution [1]Department of Artificial Intelligence, Yonsei University, Seoul, Korea. Correspondence to: Albert No <albertno@yonsei.ac.kr>.

*Proceedings of the $43^{rd}$ International Conference on Machine Learning*, Seoul, South Korea. PMLR 306, 2026. Copyright 2026 by the author(s).

knowledge" is ambiguous under entailment: if the target is "Paris is the capital of France," should the system also avoid entailed statements such as "the Eiffel Tower is in the capital of France"? Entity removal is similarly underspecified: "forget Stephen King" could refer to biographical facts, his works, quotations, or derivative discussion (e.g., adaptations). This subjectivity can make it difficult to specify a precise forget set $F$, but that is not the core issue. More fundamentally, the objective is defined by an application policy (i.e., what the model should or should not do) so the problem is inherently about policy specification and compliance rather than dataset-defined deletion.

The gap between dataset-defined deletion and policy-defined behavior control is clearest for *derived capabilities*, where training influence is not limited to memorizing the forget set (Thaker et al., 2025; Jia et al., 2025). For example, suppose a model is trained on unauthorized synthetic mathematical reasoning traces and is later required to "unlearn" them. If evaluation only checks whether the model fails to answer the same questions from that dataset, a trivial non-disclosure strategy can appear successful. The relevant question is whether the unauthorized data contributed to a *transferable* reasoning capability: if the model still solves broad classes of challenging math problems, influence may persist even when direct reproduction is blocked. Under retraining-indistinguishability, maintaining such a capability is acceptable only if the retrained model on $D \setminus F$ achieves it; otherwise, the capability should disappear along with the influence that induced it.

This terminological ambiguity also directly affects benchmarks and metrics. Many evaluations operationalize "forgetting" as output failure on a designated probe set, e.g., lower QA accuracy, lower ROUGE to a reference answer, or reduced likelihood of a target phrase (Jin et al., 2024; Yuan et al., 2025; Xu et al., 2026). Such measures can be useful diagnostics for non-disclosure, but they are not evidence of retraining equivalence. They are also often subjective (depending on QA construction and prompting context), and *lower is not always better*: a retrained model on $D \setminus F$ may still produce partially correct or contextually reasonable answers (hence non-zero ROUGE), while a blanket refusal can drive ROUGE toward zero while diverging from the retrained reference. Benchmarks therefore add retain/utility constraints (Maini et al., 2024; Shi et al., 2025; Chang & Lee, 2025), but these too encode application-dependent choices about what counts as utility and what trade-offs are acceptable. Without an explicit retrain reference, evaluation can unintentionally prioritize output control over removal of training influence.

In this position paper, we argue that resolving this confusion requires stricter terminology tied to explicit guarantees and baselines. We formalize machine unlearning as retraining-indistinguishability for a precisely defined forget set, organize other common "unlearning" usages by intent, and explain why benchmark design must reflect the distinction, especially in the presence of derived capabilities.

> **Position.** "Machine unlearning" should mean retraining indistinguishability for a precisely defined forget set; other safety- or application-driven "forgetting" goals are different problems and should use different terms.

**Conflict of Interest Disclosure.** The authors have no financial conflicts of interest to disclose.

## 2. What the Literature Calls "Unlearning": A Definition and a Taxonomy by Intent

The term *unlearning* is now used in LLM research to describe a wide range of interventions that share a surface-level motivation, making some information, behavior, or training influence "go away," but differ substantially in their intended guarantee. This terminology overload matters because different intents demand different baselines and different evaluations: a method designed to *block disclosure* can look successful under standard "forget set" tests while failing to remove the *training influence* of the forget set. In this section, we (i) give a *formal definition* of **machine unlearning** (the usage we argue should be preserved), and (ii) organize other common uses of the term into *high-level categories by intent*, without attempting rigid, mutually exclusive formalization. I.e., "unlearning" is not equivalent to making a model refuse certain questions, nor to reducing the likelihood of a particular string, nor to overwriting an answer with a replacement.

### 2.1. Unlearning: Dataset-Defined Deletion Guarantee

**Setup.** Let $D$ be the training dataset and let $F \subseteq D$ be the *forget set*, whose *training influence* is to be removed. Define the retain set as $R := D \setminus F$. Let $\mathsf{Train}(\cdot)$ denote the (randomized) training procedure, and write $\Theta_S \sim \mathsf{Train}(S)$ for the random model obtained by training on dataset $S$. An unlearning is a (possibly randomized) procedure that takes a trained model and a forget set and returns an updated model:

$$\Theta' \leftarrow \mathsf{Unlearn}(\Theta_D, F).$$

Informally, machine unlearning aims to remove the influence of training on $F$ as if the model had never seen it.

**Definition 2.1** (Exact machine unlearning (Izzo et al., 2021)). Unlearn achieves *exact machine unlearning* (with respect to Train) if for all $D$ and all $F \subseteq D$,

$$\mathcal{L}(\Theta') = \mathcal{L}(\Theta_R), \quad \text{where } \Theta_R \sim \mathsf{Train}(R),$$

and $\mathcal{L}(\cdot)$ denotes the induced distribution over model parameters (and over the randomized training outcome).

In practice, exact unlearning is a very strong requirement and is rarely attainable for large-scale models. Accordingly, most work adopts relaxed notions of unlearning that allow the unlearned model to be *approximately* indistinguishable from the retrained baseline.

**Definition 2.2** (Approximate machine unlearning (general form)). Fix a divergence/metric $\mathrm{Dist}$ between distributions and a tolerance $\tau \geq 0$. Unlearn achieves $(\mathrm{Dist}, \tau)$-*approximate machine unlearning* if for all $D$ and $F \subseteq D$,

$$\mathrm{Dist}(\mathcal{L}(\Theta'), \mathcal{L}(\Theta_R)) \leq \tau.$$

Here, $\mathrm{Dist}$ may be defined in parameter space or in *behavior space* (e.g., after composing the model with a prompt distribution and a decoding rule). We emphasize that *multiple* choices of $\mathrm{Dist}$ are reasonable; the key is that the baseline is always retraining on $D \setminus F$.

One widely used relaxation (inspired by differential privacy (Dwork, 2006)) defines closeness via $(\varepsilon, \delta)$-indistinguishability. For random variables $X$ and $Y$, write $X \approx_{\varepsilon, \delta} Y$ if for all measurable sets $S$,

$$\Pr[X \in S] \leq e^{\varepsilon} \Pr[Y \in S] + \delta$$
$$\Pr[Y \in S] \leq e^{\varepsilon} \Pr[X \in S] + \delta.$$

Then Unlearn is $(\varepsilon, \delta)$-*approximate* if $\Theta' \approx_{\varepsilon, \delta} \Theta_R$. This is a principled and popular choice, but it is *not the only* way to formalize approximate unlearning.

Interventions that change model behavior or outputs can serve many practical goals, but they are *not equivalent* to machine unlearning as defined above. The defining criterion of machine unlearning is removal of the influence of a *precisely specified* forget set $F$, operationalized as (approximate) indistinguishability from the counterfactual model retrained on $D \setminus F$ under an explicit notion of distance.

## 2.2. Other Common Uses of "Unlearning" in LLM Papers: Categories by Intent

We now summarize several common intents that are frequently labeled "unlearning" in the LLM literature. These categories are intentionally *high-level* and *not mutually exclusive*: a single system may combine multiple mechanisms (e.g., a suppression-style fine-tune plus a refusal policy), and the boundary between, say, suppression and alignment can be blurry. Our goal is not to legislate a perfect partition, but to clarify the *primary aim* that distinguishes these lines of work from machine unlearning as defined in Section 2.1.

**Output Likelihood Suppression.** Suppression-oriented approaches aim to reduce a model's tendency to generate content associated with a designated forget target. While suppression can be implemented in various ways, we focus on methods that *explicitly modify the model's likelihood over*

*selected outputs* (Welleck et al., 2020). Common instantiations include gradient-based updates such as gradient ascent (GA) (Thudi et al., 2022; Jang et al., 2023; Yao et al., 2024; Zhang et al., 2024b) and negative preference optimization (NPO) (Zhang et al., 2024a; Fan et al., 2024; Wang et al., 2025). These methods can be effective at shifting probability mass away from restricted tokens or responses under specific prompts, but they operate primarily at the level of output distributions. As a result, they do not, in general, guarantee indistinguishability from a model retrained without that data.

**Internal Representation Obfuscation.** Obfuscation-oriented works aim to make the model *unreliable or non-informative* on targeted inputs by inducing confusion, such as low-confidence or high-entropy behavior. This is commonly achieved through representation-level manipulations that distort internal activations (Li et al., 2024; Zou et al., 2024; Wuerkaixi et al., 2025; Huu-Tien et al., 2025a;b) or entropy-based objectives that encourage diffuse predictions (Yuan et al., 2025; Entesari et al., 2025; Zhai et al., 2026). Such approaches degrade answerability on selected prompts but do not match the behavior of a retrained model.

**Knowledge Editing.** Editing-based approaches aim to *recalibrate semantic associations*, modifying the model's assertions about specific entities or facts. This is typically achieved through knowledge editing techniques (Li et al., 2025; Hossain & Kagal, 2025; Jung et al., 2025) or via replacement supervision such as counterfactual fine-tuning that induces updated responses (Eldan & Russinovich, 2023; Gu et al., 2024; Scholten et al., 2025; Liu et al., 2025). These methods can effectively redirect a model's outputs toward desired answers, but they do so by overwriting behaviors rather than removing the influence of the forget set.

**Behavioral Refusal.** Refusal strategies enforce systematic non-compliance, where the model is trained to provide "I don't know" (IDK) responses to forget related queries (Maini et al., 2024; Yuan et al., 2025). Relatedly, LUNAR (Shen et al., 2025) induces coherent abstention by redirecting the internal activations of forget-set prompts toward regions that express an inability to answer. This can also be implemented with preference optimization variants (e.g., DPO (Rafailov et al., 2023)) that explicitly prefer refusal templates over forget answers. These methods primarily alter the model's *abstention behavior* on targeted prompts and are best viewed as policy-oriented alignment.

**Inference-time Interventions.** Inference-time interventions control *deployed system behavior* without modifying model parameters. Examples include guardrails and input/output filters (Thaker et al., 2024), token-level decoding constraints (Deng et al., 2025), prompt-side manipula-

tions (Pawelczyk et al., 2024; Liu et al., 2024b), and logit differencing during generation (Ji et al., 2024; Suriyakumar et al., 2025). These methods can prevent explicit disclosure of forget-related content, but because the underlying weights are unchanged, they function as external filters rather than removal of learned information.

### 2.3. Interpreting Unlearning Claims by Guarantees

Given the diversity of objectives labeled as "unlearning," a practical reading strategy is to identify the *claimed guarantee* and its *reference baseline*. Claims grounded in (approximate) indistinguishability from retraining on $D \setminus F$ correspond to machine unlearning as defined above; claims evaluated primarily by non-disclosure or behavior change without a retrain reference should be interpreted as pursuing a different objective and assessed on those stated terms.

## 3. Why Terminology Matters for LLMs Unlearning Benchmarks and Evaluation

Terminology choices directly shape benchmark design: when multiple objectives are grouped under "unlearning," evaluations tend to collapse to what is easiest to measure, often treating "forgetting" as failure to produce reference answers on a designated probe set. This section examines how common benchmarks and metrics reflect that choice, and why such scores can favor output-control solutions even when the intended claim is removal of training influence.

### 3.1. Output-Failure Metrics Dominate Current Practice

Across recent "LLM unlearning" evaluations, the most common success signal is *output failure on forget queries*: a model is deemed to have "forgotten" if it no longer reproduces the ground-truth answer for prompts related to the deletion target. Typical instantiations include: (i) surface-form similarity between generations and the ground-truth answer, (ii) embedding-based semantic similarity or cosine distance between generated outputs and the ground-truth answer, and (iii) the model-assigned probability or likelihood of the ground-truth answer under the forget prompt.

These metrics are informative for diagnosing *what the model emits* under a specific prompting protocol. However, when behavior on forget queries is interpreted in absolute terms, with differences from a reference answer taken as evidence of success, and without any comparison to a retrained model, these metrics fail to establish the defining criterion of machine unlearning: *indistinguishability from a retrained model*. Under this interpretation, methods can perform well on output-failure metrics through mechanisms that are orthogonal to influence removal. Such behaviors may be desirable for suppression or refusal objectives, but they should not be conflated with machine unlearning.

Once output-failure metrics become the de facto score, approaches explicitly designed to block forget-related responses are structurally advantaged: they optimize exactly what the benchmark counts. This creates a feedback loop where "better unlearning" often means "more reliable output suppression" under the benchmark's prompting distribution, rather than a closer approximation to the retrain-model behavior implied by machine unlearning.

### 3.2. What Leading Benchmarks Actually Measure

We summarize representative benchmarks and their scoring protocols to clarify how metric choices can blur the distinction between retraining-equivalence and output control.

**TOFU.** TOFU (Maini et al., 2024) explicitly defines a forget set and evaluates against a *retrained* reference, which is directionally consistent with the machine unlearning definition. Its core metric, *forget quality*, compares the unlearned and retrained models via a probability-based *truth ratio* (the model-assigned probability of the correct answer relative to incorrect alternatives), and assesses similarity using two-sample tests (e.g., KS tests) on the resulting distributions.

However, several subsequent works (Yuan et al., 2025; Wuerkaixi et al., 2025) apply TOFU while omitting the retrained reference and instead judge forgetting by output non-reproduction. This shifts the evaluation away from retraining equivalence toward surface-level output failure, thereby allowing output suppression to score as successful "unlearning" despite not matching the retrained model.

**MUSE.** MUSE (Shi et al., 2025) adopts a six-way evaluation framework, including criteria such as (i) no verbatim memorization, (ii) no knowledge memorization, and (iii) no privacy leakage. MUSE includes an explicit retrained baseline and evaluates privacy leakage by applying membership inference attacks (MIAs) with AUC-based metrics, directly comparing the unlearned model against the retrained model. In contrast, verbatim and knowledge memorization are operationalized via ROUGE-L scores, where lower values on forget queries are taken as evidence of success without an explicit distributional comparison to the retrained model.

**RWKU.** RWKU (Jin et al., 2024) focuses on removing real-world knowledge and stresses robustness through a broad suite of probes, including cloze and QA prompts, membership inference attacks, and adversarial elicitation. Notably, RWKU does not rely on a retrained reference model; its scores therefore reflect how a model's outputs change under the chosen probes, rather than how closely it matches a retrained baseline. Accordingly, RWKU should be viewed as assessing robustness of knowledge suppression, not machine unlearning under a retrain-equivalence.

**WMDP.** WMDP (Li et al., 2024) casts "unlearning" as reducing hazardous capabilities, operationalized as lower QA accuracy on questions related to biological, chemical, and cyber security domains. This framing does not correspond to deletion of a well-defined training subset. Instead, WMDP evaluates whether task performance is diminished on targeted domains, and thus measures capability suppression rather than retrain-equivalent machine unlearning.

### 3.3. Adversarial Evaluation Exposes the Limits of Output-Failure Scoring

The gap between output-failure metrics and actual influence removal becomes most apparent under adversarial or stress-test evaluation. Methods that appear successful under fixed prompting protocols often fail when the evaluation setting is perturbed, revealing that the target knowledge has been suppressed rather than unlearned.

Simple input-level variations (Maini et al., 2024; Lynch et al., 2024; Łucki et al., 2025; Jeung et al., 2025), such as paraphrasing, mixed queries, multilingual queries, or jailbreak prompting, are frequently sufficient to recover supposedly forgotten content. This observation is consistent with the benchmark analysis above: criteria that judge forgetting based solely on output non-reproduction do not reliably track whether the underlying training influence persists.

More revealingly, model-level interventions further expose the brittleness of output-failure-based evaluation. Prior works show that small amounts of additional fine-tuning (Yoon et al., 2026), benign post-training transformations such as quantization (Zhang et al., 2025), or targeted activation-level manipulations (Seyitoğlu et al., 2024), or representation-level auditing (Goel et al., 2026) can reveal information that was assumed to be forgotten. Such phenomena indicate that failure to emit a reference answer does not imply that the corresponding knowledge has been eliminated from the model.

## 4. Derived Capabilities: Unlearning Beyond Surface-Level Outputs

Under the machine unlearning definition, the target is the *training influence* of a precisely specified forget set $F$, not individual responses. This matters because training influence can be distributed and may persist as generalizable behaviors that extend well beyond the original examples. Consequently, output-based evaluations alone cannot determine whether the influence of $F$ has been removed.

### 4.1. What We Mean by "Derived Capabilities"

We use the term *derived capability* to denote a form of behavioral competence that is plausibly attributable to training

on $F$ (or its interaction with the rest of training), and that generalizes beyond the exact examples contained in $F$. Such capabilities need not take the form of verbatim memorization. They may instead manifest as transferable reasoning skills learned from reasoning traces, trigger-conditioned or adversarial behaviors induced by a small number of poisoned samples, persistent stylistic or tool-use habits, or latent factual competence that can be recovered under paraphrase or benign post-training interventions even when direct regurgitation is suppressed. These phenomena are conceptually important because they reveal why "unlearning = not answering" is an incomplete operationalization.

### 4.2. Implications of the Machine-Unlearning Definition

Machine unlearning is defined by (approximate) equivalence to retraining on $R = D \setminus F$. This has an unavoidable but clarifying implication:

> *If training on $F$ contributes to a derived capability, then an unlearned model that is truly indistinguishable from retraining without $F$ must also lose that capability (to the extent that the retrained model lacks it).*

This implication is often uncomfortable in LLM settings because many forget sets are entangled with desirable performance. Nevertheless, it is logically consistent with the compliance-driven unlearning objective: the correct reference is what would have happened had the model never been trained on $F$, even if that counterfactual model is less capable on some tasks.

This also highlights a terminological boundary. If an algorithm is designed to *preserve* a capability that is, in fact, attributable to $F$, then the algorithm may still be a very useful *suppression/editing/alignment* technique, but it is not solving machine unlearning as defined in Section 2.1.

### 4.3. Case: Unauthorized Synthetic Reasoning Traces

Derived capabilities are especially salient when training involves synthetic supervision. Consider a setting in which mathematical reasoning traces generated by a frontier LLM are used, without authorization, to train another model, resulting in a measurable improvement in mathematical reasoning performance. In practice, several model providers explicitly prohibit the use of their outputs to train or fine-tune competing models, and such use may later trigger a request to "unlearn" the unauthorized data.

In this scenario, the central question is not whether the model can reproduce specific solutions from the synthetic dataset. Rather, it is whether the unauthorized reasoning traces contributed to a general mathematical reasoning capability that transfers beyond the original examples.

If evaluation treats output failure on the unauthorized data as the sole success criterion, a model can appear successfully unlearned by simply refusing to answer or producing irrelevant responses, while retaining the improved reasoning ability. Such evaluation therefore fails to test whether the training influence on a transferable capability has been removed. This case illustrates why output non-reproduction is an unreliable proxy for unlearning whenever the effect of the forget set manifests as a derived capability.

### 4.4. Case: Poisoning as Derived (Adversarial) Capability

Data poisoning offers a clear example of derived capability because the intended effect is explicitly *not* memorization. Poisoned samples are crafted to induce indirect behaviors, such as degraded accuracy, targeted misclassification, or trigger-conditioned backdoors. If unlearning is equivalent to retraining without the poisoned data, then removing the forget set should also eliminate the induced behavior.

In practice, this standard is difficult to meet. Recent work (Pawelczyk et al., 2025) evaluates unlearning in settings where the forget set consists entirely of poison samples and measures whether the poisoning effects are mitigated. Across multiple attack types and model architectures, existing unlearning methods often fail to remove the induced behaviors. This makes poisoning-based evaluation particularly informative: it probes whether training influence has been removed at the behavioral level, rather than whether a model merely avoids reproducing specific outputs.

## 5. Call for Action: Reference-Based Evaluation and Derived-Capability Probes

To make progress toward the stated goal of machine unlearning, we argue that evaluation protocols should (i) anchor claims to an explicit reference model that represents "training without $F$" (or the best available approximation), and (ii) include tests for derived capabilities that capture training influence beyond surface-level non-disclosure.

### 5.1. Evaluate Unlearning Against a Reference Model

For machine unlearning, success should be judged by how closely the unlearned model matches the behavior of a reference model intended to approximate the counterfactual model trained on $D \setminus F$. The ideal reference is a retrained-from-scratch model on $D \setminus F$, but in many LLM settings this may be infeasible; in such cases, the reference should be the strongest reasonable proxy *and must be stated explicitly*.

We do not claim that output-level metrics (e.g., ROUGE, cosine similarity, forget QA accuracy) are intrinsically flawed. The issue is treating them as a *stand-alone* criterion for "unlearning" without any reference model. In that setting, score drops primarily reflect output control under a particular probe distribution, not removal of training influence.

When a paper claims machine unlearning (rather than output control objectives), we recommend reporting:

**A reference model, with provenance:** ideally $\text{Train}(D \setminus F)$ with matched hyperparameters. If this is infeasible, use the best available proxy (e.g., a matched-stage retrain for fine-tuning unlearning, a smaller-scale retraining study, or the strongest checkpoint *before* the introduction of $F$) and clearly state what it approximates and what it does not.

**Distances to the reference:** in addition to forget-query scores, report distributional/behavioral comparisons to the reference model (e.g., logit- or probability-based statistics, two-sample tests, MIA-style audits, and robustness under paraphrase and adversarial elicitation (Maini et al., 2024; Shi et al., 2025; Cho et al., 2025)), since the central question is similarity to the counterfactual baseline.

**Utility relative to the reference:** retain/utility metrics should be reported as part of the comparison to the reference model (not merely as a separate "do not break the model" constraint), since acceptable trade-offs are application-dependent and should be made explicit.

Anchoring evaluation to an explicit reference model is necessary not only for detecting memorization, but also for interpreting capability-level effects: if a capability is absent from the counterfactual reference, then retaining it after "unlearning" indicates remaining influence; if it is present in the reference, unlearning should not suppress it.

### 5.2. Derived-Capability Probes Should Be First-Class

The preceding discussion leads to a simple recommendation: when the claim is removal of training influence, evaluation must probe derived capabilities, not only reproduction of the forget set. Output-level failure on forget queries is insufficient whenever the effect of $F$ manifests as a transferable behavior. Concretely, we recommend that unlearning benchmarks include the following components:

**Capability-level holdout tasks.** Benchmarks should evaluate capabilities plausibly induced by $F$ using held-out tasks that do not reuse prompts from the forget set. Crucially, these capability-level holdout tasks are *not* utility or retain-accuracy evaluations: their purpose is not to measure general task performance, but to test whether a specific transferable capability attributable to $F$ persists beyond direct reproduction. In the synthetic reasoning case (Section 4.3), this entails testing on out-of-distribution mathematical reasoning problems rather than the original training questions.

**Intervention-based recovery tests.** Unlearning evaluations should include intervention-based recovery tests. If information or behaviors deemed "forgotten" can be recovered through benign fine-tuning (Hu et al., 2024) or other post-training transformations (Zhang et al., 2025), in contrast to a retrained reference model, this indicates that the influence of $F$ remains latent rather than removed.

**Task-appropriate threat models.** In poisoning and backdoor settings, evaluation should directly measure the induced behavior (e.g., trigger success rates, targeted error rates, or poison influence), rather than relying on failure on curated QA prompts. Such settings provide a stringent test of unlearning because the target is a derived behavior rather than surface-level regurgitation (Pawelczyk et al., 2025).

These probes are not intended to replace standard forget-query metrics. Rather, they guard against a systematic failure mode in which "unlearning" success is declared based on surface-level output suppression, while distributed and transferable training effects persist.

Finally, derived-capability probes must be interpreted *relative to the retrain reference* (Section 5.1): the goal is not to maximize forgetting signals, but to match the counterfactual behavior of training on $D \setminus F$.

# 6. Alternative Views

In this section, we describe several credible opposing views and explain where we agree, where we disagree, and what we think follows for evaluation and terminology.

## 6.1. View 1: Policy-driven removal as unlearning

**Why this view is reasonable.** Under a broad interpretation of "unlearning" as "making the model forget X," policy-driven objectives naturally fall under this label. These include preventing unsafe responses, removing or attenuating knowledge about specific entities, and rewriting particular facts. Such objectives are practically meaningful: they address safety, product policy, reputational risk, and user-facing redaction requirements, and they often admit scalable solutions that do not require access to the training data.

**Our response.** We agree that these directions are valuable and should continue to be pursued. Our concern is not with the objectives themselves, but with the interchangeable use of terminology and the guarantees it implies. Policy-driven removal is typically defined by design choices about what a model should or should not produce, which are inherently subjective and application-dependent. Even approaches that rely on structured representations such as knowledge graphs or ontology-based closures must commit to specific choices about scope, closure, and propagation, none of which are

uniquely determined by the training data (Wei et al., 2025; Luo et al., 2025).

By contrast, machine unlearning is defined with respect to an objective substrate: a precisely specified forget set $F \subset D$, with success evaluated relative to the counterfactual model retrained on $D \setminus F$. This distinction has concrete consequences for evaluation and interpretation. Using the same term for both policy-driven behavior control and dataset-defined deletion blurs whether a method claims retraining equivalence or only compliance with a policy constraint, and makes results difficult to compare across works.

## 6.2. View 2: Isn't It Enough to Not Answer?

**Why this view is reasonable.** In many deployments, the primary risk is *exposure*: users should not be able to elicit certain content such as private data, copyrighted excerpts, or unsafe instructions. From this perspective, "successful unlearning" can be viewed as an *interface-level* requirement: the system should reliably avoid producing the targeted information when queried. This interpretation is practical, aligns with product and safety objectives, and underlies many existing benchmarks (Li et al., 2024; Jin et al., 2024).

**Our response.** We agree that non-disclosure is a valuable and often sufficient objective when the goal is limited to controlling outputs. However, it is not equivalent to machine unlearning as defined in Section 2.1, nor is it sufficient when the application requires removal of training influence or robustness to post-processing.

Preventing answers on a fixed family of prompts primarily measures output control under a chosen elicitation protocol. Such control can be achieved even when the training influence of the forget set remains latent. As a result, apparent forgetting may fail under adversarial evaluation or distribution shift (Section 3.3), where the same information or behavior can be recovered.

More importantly, training influence may persist as derived capabilities (Section 4) that are not tied to reproducing specific reference strings. In these cases, a model can avoid answering forget prompts while retaining a transferable capability induced by the forget set. If the application requires that such capabilities be removed, or that forgetting be robust to benign post-processing, then a stronger notion, *indistinguishability*, is necessary.

## 6.3. View 3: A Retrain Reference Is Infeasible for Real-World Large Language Models

**Why this view is reasonable.** Retraining a frontier-scale model after removing a subset of pretraining data is often prohibitively expensive or operationally infeasible. In many settings, the original training pipeline, detailed data

provenance, or sufficient compute may be unavailable. From this perspective, defining unlearning in terms of retraining equivalence can appear impractical, motivating the use of operational criteria such as "the model no longer reveals the targeted content under our evaluation."

**Our response.** We distinguish *conceptual definitions* from *practical feasibility*. Even when exact machine unlearning is difficult or unattainable, it remains a distinct scientific and compliance-motivated objective. Replacing that objective with easier operational goals does not resolve the difficulty; it obscures what guarantee is being claimed.

We do not argue that every paper must retrain a frontier model. Rather, terminology should track the guarantee. When a retrain reference is unavailable, the appropriate response is to state and justify a proxy baseline, or to frame the contribution as addressing a different objective (e.g., non-disclosure), rather than redefining "unlearning".

### 6.4. View 4: Is a Retrained Model the Right Reference?

**Why this view is reasonable.** A natural concern is that the retrained model $\mathsf{Train}(D \setminus F)$ may not align with what stakeholders *want* a deployed system to do, even if it is the correct counterfactual for influence removal.

First, removing information can increase hallucination risk: a retrained (or successfully unlearned) model may produce plausible but incorrect answers simply because it lacks relevant evidence that was present in $F$ (Yuan et al., 2025). If a deletion request is motivated by compliance or user trust, such hallucinations may be undesirable.

Second, a retrained model may still answer "forget" questions when the same information appears elsewhere in $D \setminus F$ (e.g., duplicates, paraphrases, or independently sourced references). In that case, generating similar or even identical answers is consistent with training on authorized data, and residual similarity can persist even after removing $F$ (Jeung et al., 2026; Cooper et al., 2025). From a non-disclosure perspective, this can look like "forgetting failed," even though the deleted subset was removed.

**Our response.** We agree that retrained models can hallucinate and can still produce similar outputs, but these issues are orthogonal to the objective of machine unlearning. Machine unlearning is not a guarantee of factuality or a promise to never produce an answer; it is a claim about *removing the influence of a precisely specified dataset subset $F$*. Accordingly, (i) hallucinations caused by missing evidence are an expected consequence of removing information, and the relevant question is whether the unlearned model behaves like the counterfactual retrained model under the same conditions; and (ii) if the relevant information is present in $D \setminus F$, then both the retrained model and a correctly unlearned

model should be able to answer using authorized data.

If an application instead requires stronger behavior (e.g., abstaining from answering even when the information exists in authorized data, or enforcing strict non-disclosure regardless of counterfactual training), then the objective moves beyond "removing influence of $F$" into the domain of policy and preference design. Our recommendation is not to adjudicate which objective is preferable, but to distinguish them terminologically and evaluate them against the baseline.

### 6.5. View 5: Why Not Keep "Unlearning" as an Umbrella Term and Add Qualifiers?

**Why this view is reasonable.** The term "unlearning" is already widely used, and adopting it as an umbrella label could reduce terminological fragmentation. A natural proposal is to add qualifiers (e.g., *data unlearning* vs. *behavior unlearning*) to preserve continuity while improving clarity.

**Our response.** We view this as a reasonable compromise, and it is compatible with our intent if qualifiers are used consistently and are tied to explicit guarantees and baselines. Our concern is not the word "unlearning" itself, but the frequent mismatch between the implied guarantee and the evaluation protocol. For example, papers sometimes motivate their goal as dataset deletion or retraining-indistinguishability, yet report success primarily by driving ROUGE or forget-query accuracy toward zero, a criterion that can be satisfied by output non-disclosure and may diverge from the behavior of a model retrained on $D \setminus F$. In such cases, qualifiers alone do not prevent confusion unless they also constrain what counts as evidence.

In our view, the umbrella-term approach only works if "machine/data unlearning" is reserved for retrain-referenced deletion claims, with evaluation centered on similarity to that reference, while output-control objectives are explicitly labeled and evaluated on their own terms.

## 7. Conclusion

We argue that *machine unlearning* should be reserved for dataset-defined deletion: given a precisely specified forget set $F \subset D$, the goal is to remove its training influence by producing a model that is (approximately) indistinguishable from retraining on $D \setminus F$; other "forgetting" objectives may be valuable but require different guarantees and terminology. This distinction matters because many benchmarks reward surface-level non-disclosure on forget prompts, which can be achieved without removing influence and can miss derived capabilities; accordingly, unlearning claims should be evaluated against an explicit retraining reference (or a clearly stated proxy) and include derived-capability probes when influence removal is the goal.

## Acknowledgements

This work was supported in part by Institute of Information & communications Technology Planning & Evaluation (IITP) grant funded by the Korea government (MSIT) (No. RS-2024-00457882, AI Research Hub Project), IITP grant funded by the Korean Government (MSIT) (No. RS-2020-II201361, Artificial Intelligence Graduate School Program (Yonsei University)), and the National Research Foundation of Korea (NRF) grant funded by the Korea government (MSIT) (No. RS2025-23525649).

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

## A. Operational Objectives for Common Non-Unlearning Mechanisms

This appendix makes explicit the operational distinction behind the taxonomy in Section 2.2. Let $F$ denote the forget set and write each forget example as $(x, y) \in F$, where $y = (y_1, \ldots, y_T)$. The objectives below are representative mechanisms often called "unlearning," but they optimize output suppression, overwriting, refusal, or deployment-time control rather than similarity to the retrained counterfactual model on $D \setminus F$.

**Output likelihood suppression.** Suppression methods reduce the likelihood of outputs associated with the forget target:

$$\mathcal{L}_{\mathrm{GA}}(\Theta; F) = \mathbb{E}_{(x,y) \sim F} \left[ \sum_{t=1}^{T} \log p_{\Theta}(y_t \mid x, y_{<t}) \right]. \tag{1}$$

Minimizing Eq. (1) lowers the probability of selected forget-set continuations. This can prevent direct reproduction, but it defines success by suppressing annotated strings. The model may still retain paraphrased knowledge or derived capabilities, while a retrained model may still assign nonzero probability to similar outputs if the information remains in $D \setminus F$. Thus, suppression is an output-distribution intervention, not a deletion criterion.

**Internal representation obfuscation.** Obfuscation methods make forget-related inputs unreliable or non-informative, e.g., by inducing high-entropy predictions. Let $\mathcal{U}$ denote the uniform distribution over the vocabulary:

$$\mathcal{L}_{\mathrm{ME}}(\Theta; F) = \mathbb{E}_{(x,y) \sim F} \left[ \frac{1}{T} \sum_{t=1}^{T} \mathrm{KL}\big(p_{\Theta}(\cdot \mid x, y_{<t}) \,\|\, \mathcal{U}\big) \right]. \tag{2}$$

Minimizing Eq. (2) pushes predictions on forget-related contexts toward a diffuse distribution. This measures induced confusion, whereas retraining without $F$ need not make the model uniformly uncertain. A retrained model may answer confidently using retained evidence or fail in a structured way.

**Knowledge editing.** Editing methods overwrite the model's answer within a specified scope while preserving behavior outside that scope. Let $\bar{y}$ be a replacement response for a forget-related input $x$:

$$\mathcal{L}_{\mathrm{Edit}}(\Theta; F) = \mathbb{E}_{(x,\bar{y}) \sim F_{\mathrm{edit}}} \left[ -\sum_{t=1}^{\bar{T}} \log p_{\Theta}(\bar{y}_t \mid x, \bar{y}_{<t}) \right] + \lambda \, \mathbb{E}_{x \sim \mathcal{X}_{\mathrm{loc}}} \left[ \mathrm{Dist}(f_{\Theta}(x), f_{\Theta_0}(x)) \right]. \tag{3}$$

Here, $F_{\mathrm{edit}}$ pairs forget-related prompts with replacement targets, $\mathcal{X}_{\mathrm{loc}}$ is a locality set, and $\Theta_0$ is the original model. Editing starts from a desired replacement $\bar{y}$ and a manually chosen locality constraint. The retrained model on $D \setminus F$ need not produce $\bar{y}$, preserve the same locality set, or change only within the edited scope.

**Behavioral refusal.** Refusal methods train the model to abstain from answering forget-related queries. Let $\widetilde{y}$ denote a predefined refusal response paired with a forget-related prompt $x$:

$$\mathcal{L}_{\mathrm{IDK}}(\Theta; F) = \mathbb{E}_{(x,\widetilde{y}) \sim F} \left[ -\sum_{t=1}^{\widetilde{T}} \log p_{\Theta}(\widetilde{y}_t \mid x, \widetilde{y}_{<t}) \right]. \tag{4}$$

Minimizing Eq. (4) increases the likelihood of a fixed abstention response. This may be appropriate for non-disclosure, but it trains a refusal policy rather than approximating $\mathrm{Train}(D \setminus F)$. A retrained model may answer from retained data, answer partially, or fail without using a refusal template.

**Inference-time interventions.** Inference-time interventions modify the deployed input–output path without changing model parameters. Let $C(x) \in \{0, 1\}$ be a prompt router and $T_\sigma$ a token-, embedding-, or logit-space transformation. For an embedding-level intervention with $\mathbf{e} = E(x)$,

$$\bar{\mathbf{e}}(x) = \begin{cases} T_\sigma(E(x)), & C(x) = 1, \\ E(x), & C(x) = 0. \end{cases} \tag{5}$$

The model is evaluated on $\bar{\mathbf{e}}(x)$ for routed forget-related prompts. Any apparent forgetting depends on the router and transformation at deployment time; if they are removed, bypassed, or misroute a prompt, the original behavior can reappear. Such methods are system-level control or filtering, not removal of learned information from the model parameters.

**Operational contrast.** Equations (1)–(5) reduce forget-set performance in different ways: suppression changes likelihoods, obfuscation induces uncertainty, editing overwrites responses, refusal enforces abstention, and inference-time intervention alters the deployment path around an unchanged model. Machine unlearning is different because its reference is not a target string, refusal template, locality set, router, or transformation, but the counterfactual model trained on $D \setminus F$.

## B. Extension to the Multimodal Setting

The same terminology issue arises, and may become even more pronounced, in multimodal large language models (MLLMs). Recent work has begun to study multimodal machine unlearning and safety-oriented multimodal forgetting objectives (Huo et al., 2025; Chen et al., 2025). These directions highlight the need for clear evaluation baselines across modalities.

**Multimodal Setup.** Let $D_{\mathrm{mm}}$ denote a multimodal training set whose examples may contain text, image, audio, video, or other modality components. We write a generic example as $z = (x^{\mathcal{M}}, y)$, where $x^{\mathcal{M}}$ denotes the available multimodal input components and $y$ denotes the target text or multimodal response. Given a precisely specified multimodal forget set $F_{\mathrm{mm}} \subseteq D_{\mathrm{mm}}$, define the retain set as $R_{\mathrm{mm}} := D_{\mathrm{mm}} \setminus F_{\mathrm{mm}}$. A multimodal unlearning procedure returns:

$$\Theta'_{\mathrm{mm}} \leftarrow \mathsf{Unlearn}_{\mathrm{mm}}(\Theta_{D_{\mathrm{mm}}}, F_{\mathrm{mm}}).$$

The exact and approximate machine unlearning guarantees defined in Definition 2.1 and Definition 2.2 naturally extend to this multimodal setting. Specifically, under the dataset-defined view, *exact multimodal machine unlearning* requires the updated MLLM to be distributed identically to the counterfactual model retrained on the retained multimodal data: $\mathcal{L}(\Theta'_{\mathrm{mm}}) = \mathcal{L}(\Theta_{R_{\mathrm{mm}}})$. More practically, *approximate multimodal unlearning* evaluates success by choosing a behavior-space or parameter-space distance Dist and requiring that $\mathrm{Dist}(\mathcal{L}(\Theta'_{\mathrm{mm}}), \mathcal{L}(\Theta_{R_{\mathrm{mm}}})) \leq \tau$.

As in the text-only setting, the essential point is not the particular choice of distance, but the reference: the baseline is training without the specified multimodal forget set.

**Why multimodality sharpens the distinction.** In MLLMs, the target of a request may be expressed in one modality but revealed or preserved through another. For example, a forget target associated with an image–text pair may affect visual recognition, textual descriptions, cross-modal retrieval, VQA behavior, or downstream safety responses. Thus, a model that no longer gives the original answer to a particular visual question may still retain related information through captions, textual entity knowledge, or cross-modal representations. Conversely, a safety method that refuses image-conditioned harmful requests may successfully enforce a deployment policy while leaving the underlying training influence unchanged.

This is exactly the distinction emphasized in the main paper. If $F_{\mathrm{mm}}$ is a concrete subset of multimodal training examples, then a machine-unlearning claim should be evaluated against the criteria in Definition 2.1 or Definition 2.2, using a retrained MLLM or the strongest feasible proxy. If the goal is instead to block unsafe multimodal behavior, suppress a visual concept, or induce refusals for certain image–text prompts, then the contribution should be described as multimodal suppression, safety alignment, or policy control unless it is tied to a retraining reference.

**Evaluation implications.** A multimodal evaluation should therefore report not only forget-query performance, but also reference-relative behavior across the relevant modalities. For deletion claims, useful probes include held-out image–text pairs, paraphrased visual questions, text-only and image-only elicitation, cross-modal retrieval or captioning prompts, and modality-transfer tests that ask whether information removed from one modality remains recoverable through another. These probes should be interpreted relative to $\Theta_{R_{\mathrm{mm}}}$: retaining a behavior that is absent from the retrained reference suggests residual influence, while suppressing a behavior that is present in the retrained reference indicates output control beyond dataset-defined deletion.

