# OpenReview forum: "Position: The Term “Machine Unlearning” Is Overused in LLMs"
_ICML.cc/2026/Position_Paper_Track — ICML 2026 Position Paper Track regular_

### Official Review · Reviewer_rCX8 · 2026-03-02

**Significance:** 4
**Argument Clarity:** 3
**Rating:** 5
**Confidence:** 4

**Questions:**

See weakness.

**Alternative Views Section:**

Yes

**Compliance With Llm Reviewing Policy A Conservative:**

Affirmed.

**Discussion Potential:**

3

**Paper Summary:**

This paper argues that the concept of Machine Unlearning is overused in existing LLM literature, and is often confused with concepts such as knowledge editing, which can affect the baselines and evaluations of different methods. Therefore, the author calls for a clearer definition of related concepts.

**Position:**

Yes

**Position In Title:**

Yes

**Related Work:**

3

**Strengths And Weaknesses:**

**Strengths**
1. The issue the author focuses on is highly meaningful. Currently, there is indeed a lot of confusion in terminology related to machine unlearning in the research of LLMs. Such discussions will promote the development of related fields.
2. The writing is excellent, making it easy for me to understand.
3. The argument presented in Section 6, "Alternative Views," is comprehensive and reasonable.

**Weakness**
1. Several other viewpoints that are easily confused with machine unlearning merit more precise definitions. For example, in Section 2.2, provide a definition similar to Definition 2.1 for each term that is prone to confusion, which will make the comparison clearer. So, would you consider providing some strict definitions for the several related concepts in Section 2.2?

2. I believe the discussion in this article can be extended to multimodal large models, and the community is currently also paying attention to multimodal machine unlearning, such as [1][2]. So, will you consider machine unlearning related to multimodal large models？

[1] Mmunlearner: Reformulating multimodal machine unlearning in the era of multimodal large language models. ACL2025

[2] Safeeraser: Enhancing safety in multimodal large language models through multimodal machine unlearning. ACL2025

**Support:**

3

---

> ### Author Rebuttal · Authors · 2026-03-30
>
> We thank the reviewer for the positive and encouraging assessment. We especially appreciate the feedback that the issue we raise is meaningful, the writing is clear, and the “Alternative Views” section provides a comprehensive and reasonable perspective.
>
>
>
>
> Below, we address your suggestions regarding the formalization of related concepts and the extension of our arguments to multimodal models.
>
>
> ---
>
>
> **1. Clearer definitions of related concepts (Section 2.2)**
>
>
>
>
> We thank the reviewer for the suggestion to formalize the related concepts in Section 2.2. While we initially avoided rigid definitions because these methods often hybridize in practice, we agree that providing explicit operational boundaries will strengthen the paper.
>
>
> In our revision, we will retain the section's structure but add brief definitions and representative objectives to each paragraph (assuming standard consistency losses on the retain set are implicitly maintained):
> - Suppression: A representative Gradient Ascent (GA) objective, where minimizing the loss corresponds to gradient ascent on the forget set likelihood:$$\mathcal{L}\_{\mathrm{GA}}(\Theta_D; F)=\mathbb{E}\_{(x,y)\sim F}\left[\sum\_{t=1}^{T} \log p\_{\Theta\_D}(y\_t | x, y\_{<t})\right]$$
> - Obfuscation: A representative maximal-entropy objective, where $\mathcal{U}$ denotes the uniform distribution over the vocabulary:$$\mathcal{L}\_{\mathrm{ME}}(\Theta\_D; F) = \mathbb{E}\_{(x,y) \sim F} \left[ \frac{1}{T} \sum\_{t=1}^{T} \mathrm{KL}(p\_{\Theta\_D}(y\_t | x, y\_{<t}) ||  \mathcal{U}) \right]$$
> - Refusal: A representative "I don't know" (IDK) objective, where $\tilde{y}$ denotes a predefined refusal string (e.g., "I cannot fulfill this request"):$$\mathcal{L}\_{\mathrm{IDK}}(\Theta\_D; F)=\mathbb{E}\_{(x,\tilde y)\sim F}\left[-\sum\_{t=1}^{T} \log p\_{\Theta\_D}(\tilde y\_t | x, \tilde y\_{<t})\right]$$
>
>
> These additions sharpen the operational contrast: true "machine unlearning" is reserved for dataset-defined deletion evaluated against retraining on $D \setminus F$ (or an explicitly justified proxy). In contrast, the formulas above represent distinct, highly useful output- or policy-control mechanisms.
>
>
> ---
>
>
> **2. Scope extension to multimodal machine unlearning**
>
>
>
>
> We thank the reviewer for pointing out this interesting and timely direction. We agree that the discussion can and should be extended to multimodal large models. The community's growing focus on multimodal machine unlearning, as highlighted by the excellent recent works suggested by the reviewer [1, 2], illustrates the broader relevance of these conceptual questions.
>
>
>
>
> At a high level, we believe the core distinction discussed in our paper, separating retraining-based deletion from safety-oriented behavioral control or output suppression, remains highly relevant in the multimodal domain. As the suggested papers demonstrate, the entanglement of information across modalities makes the need for clear evaluative baselines even more critical. While a comprehensive treatment of multimodal unlearning metrics is beyond the scope of this paper, we agree that it is a crucial next frontier for the field.
>
>
>
>
> In our revision, we will add a brief discussion to make this scope explicit. We will note that our conceptual framework can also help contextualize multimodal unlearning-style problems, and we will be sure to cite these relevant foundational works to point readers toward this growing area.

---

### Official Review · Reviewer_J5uq · 2026-03-11

**Significance:** 3
**Argument Clarity:** 3
**Rating:** 4
**Confidence:** 3

**Questions:**

- How do the authors see the establishment of practical evaluation standards with regard to "specific unlearning" in LLMs? As noted in the work, a core issue with non-data-based definitions is uncertainty in effect, but the Call to Action either requires a "reference model" (infeasible most of the time) or "the best available approximation," which is currently unclear. Based on this proposal, one could expect real "unlearning" evaluations will simply not exist in the near-term future.

**Alternative Views Section:**

Yes

**Compliance With Llm Reviewing Policy A Conservative:**

Affirmed.

**Discussion Potential:**

2

**Final Justification:**

After the rebuttal and reading the other reviewers' comments, I have updated my score to favor acceptance. My concerns are largely addressed, and the paper makes a solid argument for a discussion in the community.

**Paper Summary:**

The paper advocates for a stricter (treatment of the) definition of "unlearning" in LLMs. In particular, the authors advocate that recent work that refers to "unlearning" in an LLM context no longer follows the original dataset-based (and formal) definition of the term, but instead (often implicitly) more relaxed, policy-aligned definitions. These definitions refer to unlearning more generally as the model no longer reproducing specific forms of content, which is something that, by the nature of the content (e.g., "harmful content"), becomes hard to evaluate. More concretely, the authors criticize the fact that LLM unlearning is increasingly "output-defined" (i.e., "does the model still output X"), which makes evaluation dependent on the specific way the model is used (prompted, setup, etc.).
It is argued that this overloading conflates meaning, making it unnecessarily hard to discuss progress, and in the case of many works, does not meaningfully measure "unlearning" in the classical sense but rather distinct and brittle surface-level properties. The authors make this argument across a range of recent benchmarks and datasets that target "LLM unlearning." Importantly, seemingly "unlearned" capabilities could often be restored by quite simple procedures (in some cases, even just quantizing).
The authors argue that we require stricter separation between these two types of processes, with only the dataset-dependent one being referred to as unlearning, while others can be generally grouped by the intent of the behavior they aim to prevent in LLM outputs. Evaluations of unlearning should thereby include, where possible, clear comparisons to a reference model (distance to, and relative utility) as well as strong evaluations (distributional closeness, adversarial testing (recovery), MIA, and "derived capability" tests) of the proposed technique.

**Position:**

Yes

**Position In Title:**

Yes

**Related Work:**

2

**Strengths And Weaknesses:**

##### Strengths

- The stated position is overall well-supported, and there has been a clear trend in recent LLM literature towards "alternative" notions of unlearning that can confuse people who are not actively following the field.
- I generally enjoyed the alternative views section, and the proposed compromise at the end seems to me like a feasible solution that would be a strict improvement over the current state.

##### Weaknesses

- As the authors acknowledge in the opposite view section, doing proper counterfactual reference model evaluations can be (is) in most cases, infeasible for modern models. This is also largely the reason we got to this issue in the first place, and raises the question of whether "dataset-based unlearning" should be a fundamental future goal for the community.
- With this in mind, one can see a solid argument asking for proper evaluations (as is done in section 5), but this ultimately is more in support of the position " 'unlearning' is improperly evaluated in LLMs," which is definitely true. However, looking at the referenced works, TOFU/MUSE in their baseline actually go the distance of presenting reference models trained without their respective datapoints. While RWKU does not do this, they largely follow the guidance of section 5 for better evaluations. This leaves it unclear whether many of the proposed issues are not already partly addressed in popular works in the field (at least when reading 3.1, it felt this way).

**Support:**

3

---

> ### Author Rebuttal · Authors · 2026-03-30
>
> We thank the reviewer for their detailed and thoughtful engagement. We especially appreciate the recognition that our core position is well-supported, that the alternative-views section is useful, and that our proposed compromise is a feasible improvement over the current state.
>
> Below, we address your specific points regarding existing benchmarks, the link between terminology and evaluation, and the practical feasibility of our proposed standards.
>
> ---
>
> **1. Nuances in existing benchmarks: TOFU, MUSE, and RWKU**
>
> We agree that some influential works already move in the right direction, and we do not claim the literature has failed uniformly. However, we must carefully distinguish between benchmark designs that are partially consistent with machine unlearning and their later misuse as stand-alone forgetting scores.
>
> In our revision of Sections 3.1 and 3.2, we will clarify these distinctions:
> - TOFU: This benchmark is conceptually consistent with machine unlearning because it defines a forget set $F$ and evaluates against a retrained reference. The issue arises when the benchmark is reused in follow-up works without that reference, shifting the evaluation from retraining-equivalence to mere output non-reproduction.
> - MUSE: While MUSE goes beyond simple output failure by including a retrained baseline for privacy leakage, its verbatim and knowledge-memorization criteria still rely on non-reproduction rather than an explicit comparison to retrained behavior.
> - RWKU (and WMDP): These add much stronger robustness-oriented evaluations. However, without a retrained reference, they evaluate the robustness of suppression rather than retrain-equivalent deletion.
>
> Our concern is that these benchmarks are often taken as sufficient evidence for stronger claims than they actually support. By repeatedly relying on these incomplete metrics (which measure output suppression rather than the true removal of training influence) the field is gradually overloading and diluting the precise meaning of "machine unlearning."
>
> ---
>
> **2. Link between terminology and evaluation**
>
> We appreciate the observation that our argument may read as “LLM unlearning is improperly evaluated” rather than “the term is overused.” Our intent is to show that these two claims are inherently coupled.
>
> Terminology overload creates evaluation overload. Once distinct goals (like suppression, editing, and deletion) are all labeled “unlearning,” metrics migrate across them and begin serving as evidence for guarantees they were not designed to support. We will revise the introduction and conclusion to make this causal link explicit, ensuring the paper reads as one coherent argument about guarantee mismatch.
>
> ---
>
> **3. Feasibility**
>
> We agree that exact counterfactual retraining on $D \setminus F$ is often infeasible for modern LLMs. Our proposal is not that "real unlearning evaluations will not exist until full frontier retraining is possible," but rather that we should adopt a standard based on graded evidence.
>
> We are advocating for an evidence ladder where the strength of the claim tracks the strength of the counterfactual evidence:
>
> - Gold Standard: Retraining on $D \setminus F$ when feasible.
>
> - Strong Proxies: Matched-stage retraining for fine-tuning unlearning, smaller-scale retraining studies, or using the strongest checkpoint prior to the introduction of $F$.
>
> The key requirement is that a paper must explicitly state which rung it is on and what its proxy does and does not approximate. If no retrain-relevant reference is available, the cleaner framing is to present the result as non-disclosure, suppression, or alignment.
>
> ---
>
> **4. Operationalizing the evaluation standard**
>
> To prevent "best available approximation" from sounding vague, we will concretize Section 5 into a more operational checklist for near-term evaluations:
> Specify the reference and explicitly state why it approximates training without $F$.
> Anchor the evaluation by comparing utility and behavioral/distributional metrics relative to that reference, not just the original model.
> Include robustness tests, such as paraphrase/adversarial elicitation and MIA-style audits.
> Evaluate derived capabilities by including holdouts and recovery tests when $F$ plausibly induces transferable behavior.
>
> Finally, we completely agree that the practical difficulty of true deletion is exactly why the field arrived at this point. Our position is not that the community has behaved irrationally, but that the immense difficulty of deletion has led to grouping heterogeneous goals under one label. Separating these goals now is critical so that methods are judged against the guarantees they actually target.

---

> > ### Author Rebuttal · Reviewer_J5uq · 2026-04-01
> >
> > After reading the rebuttal and the other reviews, and based on the promised adaptations in the main paper, I have adjusted my score upward. I think, particularly, the operationalization can be a helpful guidance for the community when it comes to providing proper evaluations.

---

### Official Review · Reviewer_DEyL · 2026-03-13

**Significance:** 3
**Argument Clarity:** 3
**Rating:** 5
**Confidence:** 4

**Questions:**

If unlearning is strictly defined as "dataset-defined deletion" that is indistinguishable from retraining, how should the field categorize methods that involve the direct application or removal of hard logical rules within learning-based models, rather than the removal of specific training samples?

**Alternative Views Section:**

Yes

**Compliance With Llm Reviewing Policy A Conservative:**

Affirmed.

**Discussion Potential:**

3

**Final Justification:**

The paper proposes a good condition, which deserves to be presented on ICML, I increase my score from 4 to 5.

**Paper Summary:**

This paper claims a statement that the term "machine unlearning" is severely overused in LLM research. It calls for strictly reserving the term for "dataset-defined deletion", where the training influence of specific data is removed, to distinguish it from routine safety alignment, knowledge suppression, or refusal tasks.

**Position:**

Yes

**Position In Title:**

Yes

**Related Work:**

3

**Strengths And Weaknesses:**

Pros:

1. The paper identifies the phenomenon of concept misuse, which will significantly help standardize academic terminology and benchmarks in the field. This is important to the health of field development.

2. The proposal to introduce derived-capability for evaluation is rigorous and practically constructive.

Cons:

1. The paper relies almost entirely on literature review and conceptual reclassification, lacking visible data or empirical supports.

2.  The paper fails to adequately explore the reasonable practical overlap between safety guard requirements and actual data unlearning.

**Support:**

3

---

> ### Author Rebuttal · Authors · 2026-03-30
>
> We thank the reviewer for the constructive reading and for recognizing the value of the paper’s core contribution: clarifying concept misuse and its consequences for benchmark design. We also appreciate the reviewer’s comment that the derived-capability proposal is rigorous and practically constructive.
>
> Below, we address your specific points regarding our conceptual focus, the overlap with safety guards, and the categorization of logical rules.
>
> ---
>
> **1. Relies almost entirely on literature review and conceptual reclassification, lacking visible data or empirical support.**
>
> We appreciate the reviewer’s observation. As a position paper, our primary contribution is indeed conceptual and methodological. However, we emphasize that this conceptual confusion is not merely a semantic debate. It has direct, measurable consequences for experiments and evaluation in the literature. Because the field routinely conflates suppression with deletion, current benchmarks actively reward the wrong metrics. The empirical support in our paper is precisely the synthesis of these experimental failure modes across representative works. For example, we highlight how supposedly "unlearned" models frequently fail under targeted capability probes because the underlying data influence was never actually removed.
>
> To make this point starker, we will revise the text to explicitly map these conceptual flaws to their observed evaluation failures in the literature. This added discussion will demonstrate exactly how mismatched baselines lead to flawed experimental conclusions, grounding our conceptual claims in the tangible, empirical reality of current unlearning benchmarks.
>
> ---
>
> **2. Practical overlap between safety guard requirements and data unlearning**
>
> We completely agree that there is substantial overlap in the motivation behind safety guards and unlearning, and our manuscript does not intend to dismiss this. However, our central claim is narrower: an overlap in motivation does not equate to an identical objective.
>
> In many practical deployments, stakeholders simply want a model to refrain from disclosing certain content. In these cases, a guardrail or refusal mechanism is entirely appropriate and highly effective. However, if the explicitly claimed objective is the removal of the training influence of a specific subset $F$, the mathematically and scientifically proper baseline is retraining on $D \setminus F$ (or a clearly justified proxy), not merely output suppression. We will refine the text to explicitly highlight this "same motivation, different guarantee" distinction.
>
> ---
>
> **3. Categorizing the application or removal of hard logical rules**
>
> This is a very insightful question. Our view is that methods centered on adding or removing explicit rules should be categorized by their operative mechanisms and verifiable guarantees, rather than automatically defaulting to the "machine unlearning" label.
>
> - Inference-time rules: If the rule acts as a restriction or guardrail, we categorize this as policy enforcement or inference-time intervention.
>
> - Compiled rules: If the rule is compiled into the model's weights or behavior to overwrite or constrain outputs, it aligns much closer to model editing or alignment.
>
> An intervention should only be classified as machine unlearning if the explicit claim is that the influence of a precisely defined forget set has been removed, and the resulting system is evaluated against retraining on $D \setminus F$ (or a rigorous proxy). When the substrate of the intervention is a rule rather than a dataset subset, the default classification should be policy and control.
>
> To resolve this ambiguity, we are expanding Section 2.2 to operationalize these neighboring categories. We will introduce brief working definitions for suppression, editing, behavioral refusal, and inference-time intervention, detailing the specific guarantees each can and cannot support. While we intentionally avoid overly rigid mathematical definitions, as these categories frequently overlap in practice, we agree that actively disambiguating these terms strengthens the paper.

---

> > ### Author Rebuttal · Reviewer_DEyL · 2026-04-01
> >
> > Thank you for your response. This paper do propose a good position in unlearning community, which deserves to be published. I will my score from 4 to 5.

---

### Official Review · Reviewer_QHvU · 2026-03-25

**Significance:** 3
**Argument Clarity:** 4
**Rating:** 4
**Confidence:** 4

**Questions:**

1) How do the authors propose to programmatically define and measure a "derived capability" tied to a specific forget set without access to the computationally prohibitive retrained reference model?
2) If a proxy reference model is used, how can we guarantee that the "derived capabilities" emerge and extinguish identically across scales to properly measure the unlearning effects, given known scaling laws in LLMs?
3) Does this strict definition risk making "machine unlearning" practically irrelevant for enterprise deployments, where legal requirements for redaction/removal must be balanced against the need to avoid general utility degradation?

**Alternative Views Section:**

Yes

**Compliance With Llm Reviewing Policy A Conservative:**

Affirmed.

**Discussion Potential:**

4

**Final Justification:**

I appreciate the author's constructive discussion and clarifications. I hope the author will incorporate the relevant suggestions and fulfill their commitments in the revised version. I will maintain my positive rating.

**Paper Summary:**

This position paper argues that "machine unlearning" has become overloaded in LLM research, often misapplied to policy-driven objectives. The authors advocate for a return to its rigorous origin: removing a specific forget set so the model is indistinguishable from one retrained from scratch. Critiquing current benchmarks for focusing on surface-level outputs, they introduce "derived capabilities," asserting that transferable skills gained from the forget set must also be eliminated for true unlearning to occur.

**Position:**

Yes

**Position In Title:**

Yes

**Related Work:**

4

**Strengths And Weaknesses:**

Strengths:
- The paper provides a clarification of the distinct motivations that are often mistakenly grouped under the "unlearning" umbrella.
- The authors correctly observe that current evaluation metrics, such as simple drops in QA accuracy or standard MIA, often reward superficial non-disclosure or refusals instead of actually algorithmic unlearning.
- By introducing "derived capabilities," the authors demonstrate a deep understanding of how data influences propagate through LLMs beyond verbatim memorization.

Weaknesses:
- The requirement to evaluate unlearning against a reference model retrained from scratch is computationally and operationally prohibitive for frontier LLMs. While the authors suggest using "proxies," this compromise weakens the very mathematical rigor they are arguing to preserve.
- Applying strict unlearning standards to "derived capabilities" creates an almost untestable benchmark. Because features are deeply entangled in modern LLM architectures, isolating the extent to which a general capability stems from a specific forget set versus the broader training corpus remains technically unfeasible without a full retrained reference.

**Support:**

3

---

> ### Author Rebuttal · Authors · 2026-03-30
>
> We thank the reviewer for the careful reading and the positive assessment of the paper’s clarity, discussion potential, and related-work coverage. We especially appreciate the recognition that current evaluation metrics can reward superficial non-disclosure and that “derived capabilities” matter beyond verbatim memorization.
>
> Below, we address your concerns regarding the feasibility of retraining, measuring derived capabilities, and the practical implications for enterprise deployments.
>
> ---
>
> **1. Feasibility of retraining and the role of proxies**
>
> On the feasibility of retrain-based references, our claim is conceptual before it is operational. We are not arguing that every LLM paper must retrain a frontier model. Rather, we argue that the term “machine unlearning” should track the exact guarantee being claimed. If the claim is the deletion of the influence of a specific forget set $F$, then the scientifically correct reference is retraining on $D \setminus F$.
>
> When this is infeasible, our proposal is to use the strongest available proxy and state explicitly what guarantee it does and does not support. Even if the best available reference is imperfect, making that reference explicit (and acknowledging what it actually approximates) yields a far more interpretable evaluation than simply treating output non-reproduction as evidence of retraining equivalence.
>
> In the revision, we will make this hierarchy of evidence more explicit (e.g., matched-stage retrain for fine-tuning unlearning, smaller-scale retraining studies, or the strongest checkpoint before the introduction of $F$). We will emphasize that while proxy-based evidence is necessarily weaker than full counterfactual retraining, establishing this explicit baseline is scientifically necessary to prevent the inflation of unlearning claims.
>
> ---
>
> **2. Defining and measuring derived capabilities**
>
> On “derived capabilities,” we do not mean an untestable latent essence. We mean a capability plausibly induced by $F$ that generalizes beyond direct regurgitation. The point is not to require perfect causal attribution in every case, but to move evaluation beyond “the model no longer outputs answer $X$ under prompt family $Y$.” Programmatically, the capability probe should be tied to the hypothesized mechanism of $F$. For example, if $F$ consists of synthetic reasoning traces, the relevant probe is held-out reasoning tasks outside $F$; if $F$ consists of poison samples, the probe is the induced trigger behavior or targeted error rate. In other words, the proposal is not “prove exactly how much of a capability came from $F$ without a reference,” but “test whether the behavioral effect plausibly induced by $F$ persists, and interpret that evidence relative to the best available reference.” We will clarify this distinction between exact guarantee and practical evidence more directly.
>
> We also agree with the reviewer that scale mismatch is a real concern. We do not claim a proxy can guarantee that capabilities emerge/extinguish identically across scales. Our point is precisely that, absent a retrained reference, one should not silently upgrade a weaker proxy into a strong unlearning claim. In revision, we will state more explicitly that proxy references support only proxy-strength conclusions; they are useful for calibration and partial evidence, not for asserting full retraining-equivalence.
>
> ---
>
> **3. Enterprise relevance and terminology**
>
> We do not believe the strict definition makes the topic practically irrelevant. Rather, it separates two practically important objectives that are currently conflated. Many enterprise requirements are indeed non-disclosure, redaction, or policy enforcement, and we agree these are valuable and often sufficient. Our argument is simply that such objectives should be labeled and evaluated as suppression/alignment/guardrailing when they do not claim deletion-equivalence. This distinction is practically useful because it makes the tradeoff explicit: if the requirement is “do not reveal under our interface,” then a policy method may be appropriate; if the requirement is “remove training influence of this data subset,” then the stronger unlearning objective is the right one, even if harder.
>
> We will revise the paper to sharpen these points, especially by (i) adding a clearer hierarchy of reference models and evidence strength, and (ii) clarifying that derived-capability probes are intended as first-class evidence for influence removal, not as a demand for perfect causal isolation.

---

> > ### Author Rebuttal · Reviewer_QHvU · 2026-04-05
> >
> > I thank the authors for the clarification. I agree that the term "machine unlearning" is overloaded and that the paper effectively diagnoses a major flaw in current evaluation standards. However, my primary concern remains unresolved: there is still no clean, practical operational path to actually achieve or measure true "dataset-defined deletion" at scale. Because the rigorous theoretical bar proposed is practically unattainable, and the fallback proxies inherently weaken the mathematical rigor, the practical contribution of this work is also weakened. While I believe this paper initiates necessary debate on epistemic honesty within the community, I will be maintaining my current score due to these practical operational limitations.

---

> > > ### Author Response · Authors · 2026-04-06
> > >
> > > Thank you for this thoughtful comment. We completely agree that a clean and scalable operational path to certified, dataset-defined deletion in LLMs remains an open challenge. Consequently, our contribution is not a turnkey solution, but a rigorous framework demanding clear and correct statements about how deletion claims are interpreted and evaluated. We propose that when exact retraining is not possible, researchers can still use the strongest available proxy (such as a previous version of the LLM before the targeted data was introduced) provided its limitations are explicitly stated and the resulting claims are strictly calibrated to that proxy. This distinction is practically important: it fosters transparent, meaningful progress while preventing valuable suppression techniques from being overstated as true unlearning.

---

### Decision · Program_Chairs · 2026-04-30

**Decision:**

Accept (regular)

**Comment:**

This position paper addresses a timely and significant issue within the unlearning community: the term “unlearning” is overused, leading to reduced rigor and confusion in the community, such as misuse of benchmarks, causing incorrect or insufficiently-substantiated claims. All reviewers are positive towards the paper, finding the position meaningful, well-supported, the writing quality excellent, the discussion of alternative views reasonable, and that the proposed direction can improve the state of the field.
The primary concern raised by Reviewers QHvU and J5uq was whether evaluation according to the rigorous definition of unlearning is possible in practice, given the costs of retraining LLMs. However, I agree with the authors' rebuttal: we must disentangle the practical difficulty of measuring a quantity from the definition of problem setting itself. Evaluation of machine unlearning is itself an open challenge (outside the scope of this paper, though the authors offer some guidelines such as comparing against the strongest reference point, discussing transparently the drawbacks), but this practical issue should not lead to the term being misused and to different problem formulations to become entangled.
In summary, this paper meets the track's criteria by offering an important and well-supported position that has the potential to improve the state of the field.